# Heterotrophic Plate Count Can Predict the Presence of *Legionella* spp. in Cooling Towers

**DOI:** 10.3390/pathogens12030466

**Published:** 2023-03-16

**Authors:** Marta Sanchis, Isabel Inza, Maria José Figueras

**Affiliations:** 1Serra Húnter Fellow, Facultad de Medicina y Ciencias de la Salud, Departamento de Ciencias Médicas Básicas, Unidad de Microbiología y Microbiología Ambiental, Universidad Rovira i Virgili, 43003 Reus, Spain; 2Department of Basic Medical Sciences, University Rovira i Virgili, 43201 Reus, Spain; 3Pere Virgili Institute for Health Research (IISPV), 43204 Reus, Spain; 4University Institute for Research in Sustainability, Climate Change and Energy Transition (IU-RESCAT), 43480 Vila-seca, Spain

**Keywords:** *Legionella* spp. (Lsp), heterotrophic plate count (HPC), cooling towers (CTs)

## Abstract

*Legionella pneumophila* (Lp) colonizes aquatic environments and is a potential pathogen to humans, causing outbreaks of Legionnaire’s disease. It is mainly associated with contaminated cooling towers (CTs). Several regulations, including Spanish legislation (Sl), have introduced the analysis of heterotrophic plate count (HPC) bacteria and *Legionella* spp. (Lsp) in management plans to prevent and control *Legionella* outbreaks from CTs. The 2003 Sl for CTs (RD 865/2003) considered that concentrations of HPC bacteria ≤10,000 cfu/mL and of Lsp ≤100 cfu/L are safe; therefore, no action is required, whereas management actions should be implemented above these standards. We have investigated to what extent the proposed standard for HPC bacteria is useful to predict the presence of Lsp in cooling waters. For this, we analyzed Lsp and HPC concentrations, water temperature, and the levels of chlorine in 1376 water samples from 17 CTs. The results showed that in the 1138 water samples negative for *Legionella* spp. (LN), the HPC geometric mean was significantly lower (83 cfu/mL, *p* < 0.05) than in the positive Lsp. samples (135 cfu/mL). Of the 238 (17.3%) LP samples, 88.4% (210/238) were associated with values of HPC ≤10,000 cfu/mL and most of them showed HPC concentrations ≤100 (53.7%). In addition, a relatively low percentage of LP (28/238, 11.6%) samples were associated with HPC bacteria concentrations >10,000 cfu/mL, indicating that this standard does not predict the colonization risk for *Legionella* in the CTs studied. The present study has demonstrated that a threshold concentration ≤100 cfu/mL of HPC bacteria could better predict the higher concentration of *Legionella* in CTs, which will aid in preventing possible outbreaks.

## 1. Introduction

The genus *Legionella* encompasses an increasing number of species that evolved quickly from 50 to more than 65 species at present [1,2,3]. The species are ubiquitous in aquatic habitats and water distribution systems, with the *Legionella pneumophila* serogroup 1 (Lps1) being the most important pathogenic species causing Legionnaire’s disease (LD), which is a serious pneumonia that can be fatal [1,2,3]. These bacteria grow and multiply in water at temperatures between 20 and 45 °C, with an optimal temperature range of 32–42 °C, and can grow in biofilms and replicate within eukaryotic cells such as amoeba and alveolar macrophages [1,4,5,6]. The primary route of Lps1 transmission is through the inhalation or microaspiration of contaminated aerosols containing the bacteria, which can be generated by showers and especially by cooling towers (CTs), which generate aerosols that can spread over distances as long as 0.8 to 12 km [5,7,8]. The first LD outbreak due to CTs was detected in a hotel in Philadelphia in 1976, which is when the disease was first discovered [9]. Since then, many LD outbreaks have been associated with CTs, mainly in industrialized countries worldwide including Spain, France, Italy, the United Kingdom, Norway, Australia, Germany, and countries in North America, among others [5,8,10,11,12,13,14,15,16,17,18]. Due to the increase in legionellosis cases, public health authorities have introduced recommendations, mandatory guidelines, management plans, and legislation to prevent and control outbreaks from CTs [1,13,18,19,20,21,22]. The frequent presence of *Legionella* in CTs is considered an important risk factor for local outbreaks worldwide [5,16]. As a matter of fact, in Spain in 2001, the largest outbreak of legionellosis that affected ca. 500 persons occurred in the city of Murcia [5,11]. This resulted in the immediate introduction of the first Spanish legislation in 2001, which was replaced in 2003 by a new one (RD 865/2003) that required the implementation of an auto-control management plan to prevent outbreaks due to the presence of *Legionella* spp. above certain concentrations. The principles of the Hazard Analysis Critical Control Point (HACCP) have been adopted for water installations as a safety management system under the name of the Water Safety Plan (WSP) introduced by the WHO [23,24]. The auto-control management system for CTs requires verification through the monthly analysis of heterotrophic plate count (HPC) bacteria and quarterly analysis of *Legionella* spp. (Lsp), for which thresholds for action were established, i.e., no action is needed if results for HPC are ≤10,000 cfu/mL and those for Lsp are ≤100 cfu/L (RD 865/2003). However, it is unclear to what extent the 10,000 cfu/mL established standard for HPC is adequate for predicting the presence of Lsp. The European Working Group for *Legionella* Infections (EWGLI) developed technical guidelines that include routine sampling and testing for the presence of HPC besides *Legionella* in CTs within the management plans [22]. According to the EWGLI guidelines, concentrations of HPC bacteria ≤ 10,000 cfu/mL and of *Legionella* ≤ 1000 cfu/L suggest that the system is under control [22]. However, the number of studies that have investigated the role of HPC as a predictor of the presence of Lsp in CTs is limited [1,5,13,22,25,26,27,28,29], and the data available remain controversial. According to our own previous results and those of other authors, there is a need to evaluate if regulatory measures are effective in preventing outbreaks [13,16,25,26,29]. Therefore, the present study aims to establish if HPC is a good predictor of the presence and concentration of Lsp in CTs. In addition, we also investigated if the water temperature and the levels of chlorine in the water samples predicted the presence of *Legionella*. The final aim is to evaluate if the legislative measures in place are useful for limiting the presence of high concentrations of *Legionella* in the CTs studied.

## 2. Materials and Methods

### 2.1. Water Samples

A total of 1376 water samples taken from 2013 to 2020 from the routine monitoring program of 17 CTs from an industrial area of Tarragona (Spain), which followed the requirements of the 2003 Spanish legislation (RD 865/2003) described in Table 1, were included in the study. The quality requirements of the water and makeup water used in the CTs were those of drinking-water quality. Samples for the routine culturing of HPC bacteria and *Legionella* were collected in sterile containers of 1500 mL treated with sodium thiosulfate pentahydrate to neutralize any chlorine concentration (RD 865/2003). All samples were transported according to ISO 11731-2:2004 [30]. At the time of sampling, two physical–chemical parameters were measured: temperature and chlorine concentration.

### 2.2. Culture Methods and Methodology

Water samples were filtered using 0.45 µm pore membrane filters of mixed cellulose esters (Millipore SA, Molsheim, France). For *Legionella* analysis, black gridded filters were used, following ISO 11731 [31] and ISO 11731-2:2004 [30], whereas white gridded filters were used for the determination of HPC (ISO 6222-1999) [32].

To determine the concentration of *Legionella*, 0.1, 1, 10, and 1000 mL of the water samples were filtered, and immediately thereafter 20 mL of acid buffer (pH 2.2 ± 2) was added and maintained for 5 min. Then, the filters were washed with 10 mL of sterile distilled water. Finally, the membrane filters were deposited on the *Legionella*-selective agar GVPC plates (Glycine, Vancomycin, Polymyxin, and Cycloheximide, Oxoid Ltd., Basingstoke, Hampshire, UK) and were incubated at 36 ± 2 °C for 10 days following ISO 11731-2:2004 [30]. To confirm colonies as Lsp, the plates were checked every 3 days for 10 days, and the colonies (grey/white colored) suspected of being Lsp positive were subcultured simultaneously in BCYE (Buffered Charcoal Yeast Extract) agar and BCYE agar without L-cysteine (Oxoid LTD). Colonies that grew on BCYE medium with L-cysteine, but not on BCYE medium without L-cysteine, were regarded as Lsp positive (LP). The concentration of Lsp (cfu/L) for each sample was determined on day 10, considering all the colonies that showed the aforementioned characteristics, and the global concentration for all the LP samples was expressed as the geometric mean (GM).

For the analyses of HPC, 0.01, 0.1, and 1 mL of the water samples were filtered. Thereafter, filters were deposited on Plate Count Agar (PCA) and were incubated at 36 ± 2 °C for 44 ± 4 h. The results of HPC counts (cfu/mL) were expressed as the mean or GM of the results obtained.

### 2.3. Statistical Analyses

The Student’s two-tailed t-test and a chi-squared test were performed using the Statistical Package for Social Sciences (SPSS version 15.0; SPSS Inc., Chicago, IL, USA) to determine if there were any significant differences (i) between the GM of HPC obtained in the LP and LN samples; (ii) between the temperature and chlorine mean levels in the LP and LN samples; and (iii) between the percentages of LP and LN samples at the established four ranges of HPC concentration (≤100; >100 ≤1000; >1000 ≤10,000; and >10,000 cfu/mL). Differences were considered significant at a *p*-value of <0.05.

## 3. Results and Discussion

### 3.1. Results of HPC and Legionella spp. and Values of Residual Chlorine and Temperature

From 1376 samples examined from 17 cooling towers, only 238 (17.3%) were LP, and the percentage of positive samples for each individual cooling tower varied from 9.8 to 26.3% (Table 2). The GM of *Legionella* in the LP samples was 177 cfu/L (Table 2), and the concentration found in the individual samples ranged between >2 and 407,380 cfu/L. This high upper value was encountered once, and this concentration would fall within the category of high-risk established by Miller and Kenepp [33] based on the risk assessment study that they performed, which compared concentrations of *Legionella* in CTs associated with outbreaks vs. concentrations in CTs not associated with cases of LD. In a study performed by Türetgen et al. [34] on 50 CTs, most of the LP samples showed *Legionella* levels >10^6^ cfu/L that fall within the very-high-risk category defined by Miller and Kenepp [33]. In the review of outbreaks associated with CTs performed by Walser et al. [5], the concentrations in the water varied from 2×10^3^ to 10^10^ cfu/L, whereas in an outbreak that occurred in Japan and that generated the first occupational cases of *Legionella*, the concentration found in the water of the CT was 8000 cfu/L [12].

As shown in Table 2, significant differences (*p* ≤ 0.001) were observed for the GM of HPC bacteria between the LP (135 cfu/mL) and the LN (83 cfu/mL) samples. This indicated that the concentration of HPC bacteria could predict the presence of *Legionella* in the CTs studied. These results agree with those of a previous study from Greece performed by Mouchtouri et al. [13], which investigated 96 CTs and found that there were significant differences (*p* ≤ 0.001) in the median HPC concentrations between the LP (4200 cfu/mL) and LN (422 cfu/mL) samples.

Furthermore, “a fair predictive capacity” of the HPC bacteria for the positive presence of *Legionella* in potable water was found in studies performed in the USA by Duda et al. [35,36]. They described the HPC bacteria’s fair capacity to predict the presence of *Legionella* in potable water systems because only 69% of the LP samples were correctly classified when the HPC level was >100 cfu/mL [34]. The same occurred in a study performed by Bargellini et al. [4] in hot water systems in Italy, where statistical differences (*p* ≤ 0.001) were observed in the GM of HPC bacteria in water samples that were positive (192 cfu/mL) and negative (57 cfu/mL) for *Legionella*. The results of the mentioned studies, together with our findings, support the capacity of HPC bacteria to be a proxy indicator for the presence of *Legionella* in water samples.

The two physical–chemical parameters studied, the temperature and chorine concentration, presented very similar means in the positive and negative *Legionella* samples, without significant differences (*p* ≥ 0.169) (Table 2). The mean temperatures observed in the water from LN (22.4 °C) and LP (20.9 °C) samples agreed with those observed by Mouchtouri et al. [13] in Greek CTs, where no differences (*p* = 0.3) between the temperature in the LP (26.8 °C) and in the LN (25.1 °C) were found. In a study performed in Turkey, the water temperature of the CTs ranged between 4 and 30 °C, and the detection of Lsp increased with the increasing temperature [34]. The Spanish legislation for CTs and evaporative condensers (RD 865/2003) recommends that the temperature of the water should be lower than 20 °C, establishing an alert limit for temperatures above this value. In different studies, the water temperatures of CTs range between 20 °C and 50 °C [13,37]. In this range of temperatures, Lsp represents a significant public health risk because these bacteria are able to grow to high concentrations at these temperatures [1,6,38,39]. In our study, we found *Legionella* at concentrations of 405 and 18,018 cfu/L in water samples that showed very low temperatures, i.e., 8.5 and 9.3 °C, respectively (Table 2). The WHO indicates that *Legionella* can survive for long periods at low temperatures and that when the temperature increases and other conditions allow it, it can multiply to levels that may represent a public health problem [1]. In this sense, the temperature of the water was very variable in CTs that generated outbreaks, but many studies reported that outbreaks can be associated with the inadequate maintenance of the CTs [5,6,13,18,40]. This statement was confirmed in 10 of the 19 CT outbreaks investigated by Walser et al. [5]. Studies have indicated that, when Lsp colonized a cooling tower or other water system at high concentrations, the eradication of the bacteria was very difficult [5,6,40]. In our study, a single cooling tower required 20 additional analyses for *Legionella* to those required by the regular controls (quarterly, RD 865/2003, Table 1) to re-establish the control with *Legionella* results ≤100 cfu/L. On four occasions, this additional analysis was required because the results of the monthly control of HPC bacteria were above the alert concentration of 10,000 cfu/mL. The remaining 16 analyses were associated with results of *Legionella* >100 cfu/L, which required different interventions depending on the specific concentration (Table 1). As shown in Table 1, the interventions associated with the water management program included evaluating the efficacy of the biocide and doses, or applying a cleaning protocol, which all had to be repeated until *Legionella* counts were ≤100 cfu/L.

In the new Spanish legislation (RD 487/2022), a new test for *Legionella* should be performed after cleaning, and if *Legionella* is not detected or is <100 cfu/L, a new sample should be taken after one month; if the same result is obtained, the regular monitoring program can be continued [20]. It has been indicated by Buse et al. [7] that the most important aspect is to be able to reduce the concentration or the persistence of *Legionella* at high concentrations in the system. Some authors have warned that a non-detection result for *Legionella* may give managers the imprecision that the system is under control, however, it is well known that the obtained counts may have a great variability [26,41,42,43]. Previously reported data from Greek CTs show that *Legionella* colonization was positively associated with the absence of regular control plans and aged cooling tower installations and negatively associated with the use of chemical disinfection [13]. In the latter study, performed by Mouchtouri et al. [13], free chlorine levels < 0.5 mg/L were associated with *Legionella* colonization in 55.9% of the CTs studied. In our study, 75% of the CTs with LP samples had values of free chlorine > 0.5 mg/L. However, no significant differences in chlorine levels were found between the LP (1.003 mg/L) and LN samples (0.93 mg/L) in our study. It has been suggested that a continuous chlorine application in a cooling water system may create an adverse environment for *Legionella* spp., reducing the microbial diversity and enhancing the presence of *Pseudomonas,* which seem to act as antagonistic bacteria [6,44]. Persistent high concentrations of *Legionella* (10^8^ cells/L) could only be reduced by simultaneously adding free chlorine and dissolved oxygen in a CT impacted by storm events that generated the presence of debris in the basin [40]. It is also very relevant to bear in mind that for chlorine to be an effective disinfectant, the pH should be maintained <8 (optimal disinfection at pH 7.2–7.8). In the study performed by Türetgen et al. [34], the highest densities of *Legionella* (>10^7^ cfu/L) appeared in water samples that had a pH of 8.8–8.9. As stated by Brigmon et al. [40], cooling tower operators tend to maintain water at a higher pH to avoid corrosion. Additionally, based on our experience, pH should be carefully monitored because an excess of organic contamination may also reduce chlorine effectiveness. It is worthwhile to note that the existence of water management plans and their efficient implementation are key factors in reducing *Legionella* outbreaks [5,11,13,14,15,16,18]. Clopper et al. [18] investigated 14 outbreaks associated with CTs, where the fatalities were also analyzed, and indicated that 81% of the fatalities were associated with CTs that did not have a water management plan implemented.

### 3.2. Relationship between HPC and Legionella Concentrations

In this study, we further explored if the HPC alert >10,000 cfu/mL, which is the standard established by the Spanish legislation and other regulations to initiate action (i.e., the immediate analysis of *Legionella*, among others, Table 1) was adequate. To evaluate this, we established four ranges of HPC concentrations (≤100; >100 ≤ 1000; >1000 ≤ 10,000; and >10,000 cfu/mL), and we associated each of them with the percentage of LP samples subdivided by the same four ranges of *Legionella* concentrations (Figure 1). As shown in Figure 1, the percentage of LP samples was inversely proportional to the concentration of HPC bacteria. The majority of LP samples (210/238, 88.4%) were associated with HPC bacteria values ≤10,000 cfu/mL, and most of them were associated with HPC values ≤100 (53.7%). In fact, of the 107 LP samples with HPC bacteria values ≤100, only one sample was associated with a *Legionella* concentration >10,000 cfu/L, and this represents less than 1% (i.e., 0.8%). *Legionella* concentration >10,000 cfu/L is the threshold at which, according to the Spanish Legislation, the CT should be shut down, thoroughly cleaned, and disinfected (Table 1).

As shown in Figure 1, the percentage of samples with high levels of *Legionella* increased progressively to 6.4% and to 11.4% at higher levels of HPC bacteria, i.e., ranges of >100 ≤ 1000 cfu/mL and >1000 ≤10,000 cfu/mL, respectively. However, the percentage of samples with concentrations of *Legionella* >10,000 cfu/L was relatively similar; these were 11.4 vs. 10.7% (*p* > 0.05) in the samples with HPC bacteria in the ranges of >1000 ≤10,000 cfu/mL and >10,000 cfu/L, respectively (Figure 1). These apparently incongruent results could be associated with the inhibitory effect of the overgrowth of competing microbiota on *Legionella*, which has been described by different authors as one of the important limitations of conventional culture methods [2,7,29,45,46]. It has been recognized that there exists a great uncertainty associated with the *Legionella* cfu counts obtained by the current ISO 11731 gold standard method, which recommends different processing culture methods depending on the type of sample expected, i.e., clean or contaminated [2,7,29,45]. Heterogeneity of results may be due to many factors, and the differences can be generated by the procedures, i.e., the direct plaiting vs. the membrane filtration approach, the acid treatment, the elution of the bacteria from the filters, the inhibitory growth due to interfering bacteria, dilution errors, etc. [2,7,29,45]. A comparison between the ISO 11731-2017 and a quantitative PCR (qPCR) method concluded that the latter was fast and reliable, especially for samples with high levels of interfering microorganisms, and it was considered an excellent and fast screening and monitoring tool for identifying LN and LP samples [47]. Apart from the aforementioned uncertainties, we must also consider the fact that *Legionella* can be found in different forms, i.e., planktonic, associated with amoeba cyst, and biofilm [6,7].

In conclusion, based on the results obtained in our study, the limit of HPC ≤ 100 cfu/mL quite accurately predicts the absence of a higher concentration of *Legionella* in the CTs (>10,000 cfu/L), in contrast to the >10,000 cfu/mL threshold proposed by the Spanish legislation. In addition, these results clearly indicate that this threshold does not predict to any extent the colonization risk for high concentrations of *Legionella* in the CTs studied.

Furthermore, we explored to what extent HPC results ≤ 10,000 cfu/mL that required no action were associated with concentrations of Lsp >100 cfu/L (Table 3) which according to the management protocol would have required different interventions depending on the specific concentration (Table 1). The results in Table 3 reveal that almost 50% (115/238, 48.3%) of the LP samples showed *Legionella* concentrations >100 cfu/L with HPC ≤ 10.000 cfu/mL. Therefore, the HPC standard was not useful for predicting levels of *Legionella* that required action and were unnoticed. As a matter of fact, of the 11 (4.6%) LP samples that showed a concentration of *Legionella* higher than 10,000 cfu/L, 8 (73%) exhibit HPC concentrations <10,000 cfu/mL. This again questions the value of this standard in signaling the presence of high concentrations of *Legionella*, which could represent an important public health problem. 

It has been indicated that the most important aspect when trying to control *Legionella* in water systems is the reduction of its persistence at high concentrations [7]. In this sense, outbreaks attributed to cooling towers have been associated with a concentration of 8000 cfu/L [12], or between 10,000 and 5.5 × 10^6^ cfu/L [48]. Shelton et al. [48] compared concentrations found in CTs associated with outbreaks with those found in CTs not associated with LD, and they suggested that CTs with very high colony counts may be substantially more likely to be the source of outbreaks than cooling towers with lower counts.

Several countries have proposed guidelines and regulations where HPC was included in parallel to the study of *Legionella*. For instance, Europe’s Industry Association for Indoor Climate, Process Cooling, and Food Cold Chain Technologies [49] recommends values of <10,000 cfu/mL for HPC and <1000 cfu/L for *Legionella* and only recommends increased biocide treatment if HPC counts are between 10,000 and 100,000 cfu/mL. The measurement of *Legionella* counts is only recommended if high HPC counts persist. Other guidelines that recommend values for HPC and *Legionella* for water systems and commercial buildings are those proposed by the following: i) The Association of Water Technologies (AWT), <1000 cfu/L for *Legionella*; ii) The UK Health and Safety Executive (HSE), <10,000 cfu/mL for HPC and ND or <100 cfu/L for *Legionella*; iii) Joint Australian/New Zealand standards (AS/NZS 3666.3:2011), <100,000 cfu/mL for HPC and no detection or <10 cfu/L for *Legionella*; and iv) EWGLI, <10,000 cfu/mL for HPC and <10,000 cfu/L for *Legionella*, below these levels no action is required [50,51,52,53].

Even if the concentration of HPC is not directly correlated with the concentration of *Legionella* in cooling tower samples, such high levels of HPC (>100,000 cfu/mL) could indicate that the system could be out of control and contain *Legionella*, according to the Public Health Division of Melbourne [54].

Many authors and guidelines indicate that the risk of an outbreak of *Legionella* occurring in a cooling tower is not only dependent on the HPC counts, but that there are many factors that contribute to the presence and replication of *Legionella*, i.e., temperature, operation, season, presence of protozoa, etc. [4,5,6,26]. Microbiome studies of cooling tower ecosystems revealed that the composition of bacteria, ciliate, and amoeba species might interact positively or negatively with *Legionella* [44,55]. Paranjape et al. [44] suggested that the cooling tower microbiome could be manipulated to generate an ecosystem that could not be colonized by *Legionella.*

Our results agree with other studies, showing that hot water samples or drinking water in which HPC concentrations were ≤100 cfu/mL had significant associations with *Legionella*-positive contamination [6,56,57]. Bargellini et al. [6] studied 408 hot water samples from different public and private systems applying the Italian guidelines that indicated that levels of *Legionella* above 10,000 cfu/L require interventions. In the study, they found 194 positive *Legionella* samples (GM concentration of 4500 cfu/L) and observed that the cut-off of HPC ≤27 cfu/mL (22 °C) and ≤150 cfu/mL (37 °C) could be a good predictor for the presence of *Legionella* [6]. In addition, in a study performed by Glažar Ivče et al. [58] on drinking water distribution systems in Croatia, they found that HPC at 37 °C and HPC at 22 °C showed significant positive correlations with the *Legionella* counts. Similar results were also found by Solimini et al. [59] that indicated the need for controlling HPC in water systems of buildings used by individuals who may be at particular risk of being infected by *Legionella.* In fact, the new European Directive EU 2020/2184 for drinking water (DW) introduces *Legionella* enumeration as a new microbiological parameter relevant to the risk assessment of domestic distribution systems, with a parametric value of <1000 cfu/L to initiate action [60]. However, it indicates that action can be taken below this value, e.g., in cases of infection and outbreaks. In such cases, the source of the infection should be confirmed, and the species of *Legionella* should be identified. This directive has recently been adopted into the Spanish legislation (RD 3/2023), introducing the non-compliance parametric values for HPC and *Legionella* of <100 cfu/mL and <1000 cfu/L, respectively [61]. These reference values agree with the results of the current and previous studies [25] and will probably contribute to lower disease outbreaks. A study performed by Zang and Lu [62] also encourages the introduction of *Legionella* as an additional parameter in the DW legislation of the United States.

As stated earlier, the new Spanish legislation for the prevention and control of legionellosis (RD 487/2022) that will have to be implemented from 2023 onward, introduces changes in the existing threshold for the HPC standard, increasing the value from >10,000 cfu/mL to ≤100,000 cfu/mL, and extends the sampling regime (from monthly to every three months). From our point of view, both the change in frequency of HPC bacteria sampling and the new HPC standard will make this parameter completely irrelevant, and its predictive capacity for *Legionella*, which was the reason for using it in the former legislation, will be totally lost. In effect, the value of the HPC bacteria as an indicator of the number of interfering bacteria in the water system relies on the fact that their analysis is very cheap and fast (it provides results in 48–72 h) when compared with the conventional culture method for *Legionella*, which is much more expensive and time-consuming (it requires 10 days). An alternative most probable number method provided results in 7 days [63]. Molecular methods (qPCR) fulfill the characteristic of being fast, i.e., provide results within the same day, but are also not free of inhibition problems [29,45]. In addition, they do not allow the recovery of the bacteria for comparison in the case of suspected outbreaks. In the new Spanish legislation, the frequency of *Legionella* analysis changes from quarterly to monthly (Table 1). This can be considered an important gain because more frequent data on *Legionella* concentrations added to the system will allow for much sooner initiation of corrections. In contrast, this change will considerably increase the cost of monitoring and the work generated for the laboratories. The new requirement answers the claim that the determination of the health risk associated with CTs cannot be based upon a single or infrequent *Legionella* test because sporadic no-detection results would give the idea that the system is under control, when in reality it is not [6,26,41,43,53].

In conclusion, considering the low number of positive *Legionella* samples found in the CTs studied our study (17.35%), we should indicate that the mandatory controls and strategies included in the Spanish legislation (RD 865/2003) have been quite effective in reducing the presence of this bacteria in the cooling water systems studied. However, the present study has demonstrated that levels of HPC ≤100 cfu/mL could better predict higher concentrations of *Legionella* and prevent possible outbreaks. Applying an HPC value of ≤100 cfu/mL as the standard for predicting the presence of *Legionella* >10,000 cfu/L failed only in a single sample (0.8%), and this represents a very high level of precision. In fact, the failure in this prediction increases progressively when HPC bacteria were >100 cfu/mL. Therefore, the <10,000 cfu/mL HPC threshold for initiating action, established in the Spanish legislation, other legislations, and by EWGLI, does not seem appropriate based on our results. For this reason, HPC ≤100 cfu/mL could be a better early warning system for increasing concentrations of Lsp in a system that could be at risk of generating outbreaks. We agree with many authors, who indicate that faster and more reliable methods to support the management actions are needed, along with the implementation of WSP to prevent outbreaks due to the presence of *Legionella* spp.

## Figures and Tables

**Figure 1 pathogens-12-00466-f001:**
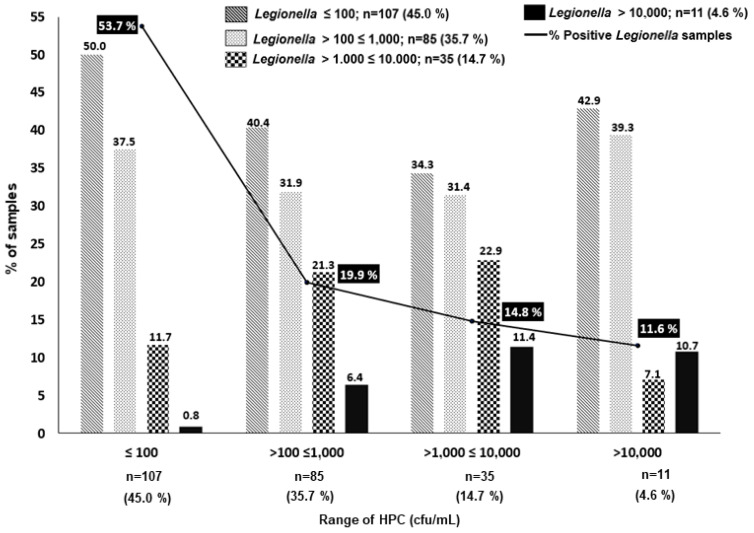
Percentage and concentration of the 238 *Legionella*-positive samples at different ranges of HPC bacteria. Note that only 28 samples showed HPC > 10,000 cfu/mL.

**Table 1 pathogens-12-00466-t001:** Microbiological analytical requirements of the Spanish legislation (RD 865/2003) and management actions associated with the control of *Legionella* in cooling towers and evaporative condensers.

Microbiological Parameter/Frequency	Threshold	Interventions
HPC/monthly *	≤10,000 cfu/mL	Continue with the management plan
	>10,000 cfu/mL	Check the efficacy of disinfection (doses and biocide) and perform immediate sampling for *Legionella* analysis
*Legionella* spp./quarterly *	≤100 cfu/L	Continue with the management plan
	>100 cfu/L	>100 ≤1000 cfu/L review the management plan, introducing corrections if needed, and take new samples after 15 days * (until *Legionella* levels are ≤100 cfu/L)
		>1000 ≤10,000 cfu/L review the maintenance program, perform cleaning and disinfection, and resample as above
		>10,000 cfu/L shut down the cooling tower

* New Spanish legislation (RD 487/2022) establishes the need for interventions if results for *Legionella* are ≥100 cfu/L, changes the frequency of analysis to monthly, and establishes that resampling should be done after 15 and 30 days. It also changes the level for HPC to ≤100,000 cfu/mL and the frequency to quarterly. In our study, only two LP samples showed levels of HPC above 100,000 cfu/mL, with *Legionella* concentrations of 18 and 69 cfu/L, respectively.

**Table 2 pathogens-12-00466-t002:** Number (%) of *Legionella* negative (LN) and positive (LP) samples associated with heterotrophic plate count (HPC) bacteria and values of chlorine and temperature.

Type of Samples (n/%)	HPC Positive Samples (%)	Geometric Mean (GM)		Free-Residual Chlorine (Mean mg/L)	Mean Temperature (Range)
		HPC (cfu/mL)	*Legionella* (cfu/L)		
*Legionella* negative (LN) (1138/82.7)	95.1 %	83 *	─	1.003	22.4 °C (2–40)
*Legionella* positive (LP) (238/17.3)	100 %	135 *	177	0.93	20.9 °C (5–35) **

* Significant differences (*p* < 0.05) between the GM of HPC in the LN and LP samples; 16 LN samples showed HPC values >100,000 cfu/mL and the concentrations range was 1 × 10^5^–1.3 × 10^6^ cfu/mL while only two LP samples had levels of HPC bacteria >100,000 cfu/mL. The percentage of positivity for each individual cooling tower ranged between 9.8 and 26.3 %. ** Four water samples showed the lowest temperatures i.e., 5.0 °C, 7.7 °C, 8.5 °C and 9.3 °C and the concentrations of *Legionella* were 106 cfu/L, 9 cfu/L, 405 cfu/L and 18,018 cfu/L, respectively.

**Table 3 pathogens-12-00466-t003:** Concentrations of *Legionella* (cfu/L) at four different ranges of HPC bacteria (cfu/mL).

Range HPC (cfu/mL)	Number of Samples (%)	Number (%) of *Legionella*-Positive Samples Distributed by Their Concentration Range (cfu/L)				
		≤ 100	>100 ≤1000 *n* = 74	>1000 ≤10,000 *n* = 33	>10,000 *n* = 8	Total 115/238 (48.3)
**≤ 100**	128 (53.7)	64 (50.0)	48 (37.5)	15 (11.7)	1 (0.8)	**64 (55.7)**
**100 ≤ 1000**	47 (19.9)	19 (40.4)	15 (31.9)	10 (21.3)	3 (6.4)	**28 (24.3)**
**>1000 ≤ 10,000**	35 (14.8)	12 (34.3)	11 (31.4)	8 (22.9)	4 (11.4)	**23 (20.0)**
**>10,000**	28 (11.6)	12 (42.9)	11 (39.3)	2 (7.1)	3 (10.7)	
**Total (%)**	**238 (100)**	**107 (45.0)**	**85 (35.7)**	**35 (14.7)**	**11 (4.6)**	

## Data Availability

There is not a public additional dataset. The data obtained are all described in the study.

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
