# Peer review of "Heterotrophic Plate Count Can Predict the Presence of Legionella spp. in Cooling Towers"

_pathogens, 2023, doi:10.3390/pathogens12030466_

Round 1
Reviewer 1 Report
The authors present the results of their study in which they compared concentrations of Legionella against those of HPC in water samples received from cooling towers. The current spanish legislation indicates that a number of HPC more than 10000 cfu/ml should be used as a warning tool for the potential presence of Legionella in a water system. The authors concluded that this number is much higher than what could be trully used as a warning tool. Based on their results, statistically significant results were computed at HPC numbers of more than 100 cfu/ml. It is interesting though that according to the authors, the spanish legislation will change from this less strict cut-off to an even more less one of 100000 cfu/l.
The manuscript is generally well written and well presented, the conclusions of the authors are well documented and the general impact of the results are of interest to the public.
There are only bits and pieces that need to be revised, attached you will find a pdf file with some comments that the authors may find useful.

Author Response
The responses are included in the document.

Reviewer 2 Report
I think that the manuscript covers a subject of public health consern and it is also well written. Nevertheless, It has some issues that must be noticed by the authors.
General comment: It is reported that the legislation for Legionella and HPC detection (RD 865/2003), has been in place since 2003. Untill to date, it has not been amended? Because later you refer the (RD 487/2022). Why you did not follow the new legislation? When did the samplings take place? Please explain.
Abstract
“and most of them sowed HPC concentrations ≤100 (53.7 %).”Please correct “sowed”.
Introduction
The first sentence is too big. Please rephrase in two smaller ones.
“Therefore, the present study…The final aim is to evaluate if”. Please change the font, in order to matches the text.
2. Materials and methods
2.1. Water samples
It is not mentioned anywhere when the samplings took place and for how long. Please explain.
«and two physical-chemical parameters i.e., temperature and chlorine». Why i.e.? Are not measured the same parameters in every sample?
2.2. Culture methods and methodology
I noticed that you used only the acid buffer method for the Legionella detection. Why you choose only this procedure and not others, or a combination of them? Please explain.
In addition, why you incubate the dishes only at 22 ºC and not at 22 ºC and at 37 ºC? Please explain.
3. Results and Discussion 3.1. Results of HPC and Legionella spp. and values of residual chlorine and temperature
What is a “Also “a fair predictive capacity” of the HPC bacteria for the positive presence “? Please explain.
Author Response
The responses are included in the file.

Round 2
Reviewer 2 Report
I believe that the authors made all the corrections that were suggested to them. The only observation I have to make is about the incubation temperature of the dishes . It seems a little strange to me that while everywhere in the original text it is written that it was done at 22 ºC, they finally tell us that it was done at 36 ºC. Ultimately, I believe that the manuscript is almost ready to be published.
Author Response
The responses for the reviewer are attached word.
In addition, we have corrects some typographical errors that are now indicated in blue. We have unified also the reference of the ISO methods.
